# Genomic Analysis of *Staphylococcus aureus* Strains Originating from Hungarian Rabbit Farms Reinforce the Clonal Origin of Various Virulence Types

**DOI:** 10.3390/ani10071128

**Published:** 2020-07-02

**Authors:** Zoltán Német, Ervin Albert, Ádám Dán, Gyula Balka, Áron Szenes, Rita Sipos, Szabolcs Bódizs, Imre Biksi

**Affiliations:** 1Department of Pathology, University of Veterinary Medicine Budapest, István u. 2., H-1078 Budapest, Hungary; albert.ervin@univet.hu (E.A.); dan.adam@univet.hu (Á.D.); balka.gyula@univet.hu (G.B.); szenes.aron@univet.hu (Á.S.); szab.bodizs@gmail.com (S.B.); biksi.imre@univet.hu (I.B.); 2Biomi Ltd., Szent-Györgyi Albert út 4, H-2100 Gödöllő, Hungary; sipos.rita@biomi.hu

**Keywords:** *Staphylococcus aureus*, rabbits, whole genome, wgMLST, *spa*-typing, MLST

## Abstract

**Simple Summary:**

Staphylococcosis is a major disease in both human and veterinary medicine. In commercial rabbit production, highly virulent variants cause significant economic losses. This study describes the comparative analysis of the whole genome of 51 strains sequenced by our group, and 12 sequences derived from online repositories. The investigation was based on whole-genome sequencing and whole-genome multilocus sequence typing. The typical highly virulent strains showed great similarity with ST121/t645 isolates from Italy, Spain and the UK. The most prevalent genotype in Hungary was an atypical highly virulent variant; these strains form a novel sequence type (ST5993), and belong to three different *spa* types (t4770, t711 and t2407). Low virulence (LV) strains grouped into two main clusters, two ST1 LV strains formed a separate cluster from the majority forming a clone-like group, despite the five different multilocus sequence typing (MLST) patterns and seven different *spa* types. Some strains in this survey showed genetic polymorphism on more than 50% of the examined loci, however, even within the same MLST ST group, hundreds of loci showed polymorphism, which could facilitate the very fine differentiation of the strains.

**Abstract:**

Staphylococcosis is one of the most important infectious diseases in rabbit medicine, especially in commercial farming. Previous studies revealed the existence of virulent variants adapted to rabbits. Typical and atypical, highly virulent as well as low virulent variants have been isolated and reported from industrial units in all major rabbit-meat-producing countries. Preceding the research focused on detecting defined nucleotide sequences, the genome of these organisms as a whole was rarely subjected to scientific investigations. The authors sequenced 51 *Staphylococcus* strains originating from industrial rabbit farms in Hungary. Another 12 draft genomes of rabbit isolates were constructed from read sequences available in digital repositories, and were compared based on whole-genome multilocus sequence typing. The clonal origin of highly virulent variants is confirmed, the strains from Hungary were closely related with the strains isolated in the UK, Italy, and Spain. Atypical highly virulent strains are the most prevalent in Hungary, they form a separate clonal cluster. The low virulent strains were genetically similar, but more heterogeneous than the highly virulent (HV) and aHV strains even by the traditional MLST typing scheme. Other “non-aureus” *Staphylococcus* species were also identified.

## 1. Introduction

Several clinical forms of *Staphylococcus aureus* infections could cause devastating diseases in rabbit farming. Production losses due to mortality, premature elimination and slaughterhouse condemnations can result in a significant decrease in profitability. The pathogen can damage the skin and internal organs; usually subacute-chronic, purulent-necrotic lesions develop in infected animals. Reservoir lesions, e.g., mastitis or sore hocks (pododermatitis ulcerosa) in breeding animals are unable to completely heal. The lasting presence of the pathogen often results in the infection of new-born rabbits, and the consequent septicaemia and mortality seriously hinder production [1]. The contamination of farms with virulent variants of *Staphylococcus aureus* often calls for radical solutions, usually a complete depopulation and disinfection of the unit [2]. *Staphylococcus aureus* is a zoonotic agent, and a recent study reported that the rabbits and the workers on a rabbit farm can carry identical clonal types of this pathogenic microorganism [3]. Intensively producing commercial farms usually focus on production and buy young females from genetic centres. The fertility of females decreases over time, and 2–2.5-year-old, otherwise healthy females are slaughtered for meat. In order to maintain the peak production levels on the farm, usually 80–140% of the breeding flock is replaced each year. Some producers are involved in long-term contracts with suppliers, which in turn creates a stable epidemiological chain. In other cases, the farmer is looking for better production results and thus constantly changes the genetic resources. This creates a constant influx of live animals on all farms, along with the possibility of introducing new pathogenic agents to the population.

The detection of such virulent pathogens in rabbits can be accomplished by routine bacterial culturing, followed by a multiplex polymerase chain reaction (PCR) to detect three virulence determinant genes [4]. Previous studies revealed the existence of highly virulent strains (HV), which were detected in Belgium, France, Greece, Italy, Portugal and Spain [5]. Our research group previously reported the presence of such strains on Hungarian rabbit farms [6]. HV strains were identified as *Staphylococcus aureus* clonal complex (CC) type 121 (or sequence type 121, ST121), similar strains were known for their increased virulence, and were proven to be globally distributed [7]. A study revealed that a single nonsynonymous nucleotide mutation in the coding region of an integral membrane protein (*dltB)* was associated with the host adaptation of CC121 from humans to rabbits [8]. Holmes et al. (2016) reported whole-genome sequencing (WGS) results of a *Staphylococcus aureus* strain belonging to ST121, isolated from a companion rabbit in the UK [9].

Previous studies also reported atypical highly virulent strains (aHV), which were considered virulent based on the severity of the outbreak they caused in Belgium in 1994 [10]. Since two virulence genes were undetectable in these strains using multiplex PCR, this genotype was considered atypical [4]. Decades later, this genotype was found to be the most common genotype in Hungarian meat-producing rabbit populations [6].

Low virulence strains (LV) cause sporadic infections and the prevalence of such strains is difficult to estimate [2]. Previous studies found that LV strains have biochemical properties similar to the isolates of human or poultry origin [11].

Other studies on rabbit-related *Staphylococcus* isolates focused on different sequence regions [3,12,13,14,15], which proved very useful, but only a handful of genes were used to identify and classify different variants of the pathogen. Minor differences in the targeted loci can show the same distinctness as would be measured with profoundly diverse strains, and the mutations of distant variants of a species can also lead to the evolution of the same nucleotide sequence [16].

Whole-genome short read shotgun sequencing of small genomes has become an affordable method on the Illumina platform nowadays. The cost of sequencing the 2.8 Mbp genome of *Staphylococcus aureus* with high coverage is comparable to 8–10 polymerase chain reactions (PCR) and the following Sanger sequencing, which is needed for multilocus sequence typing (MLST) or staphylococcal protein A (*spa*) typing. These methods determine meticulously selected markers suitable for genotyping, but only approximately 0.1% of the whole nucleotide sequence of the bacterium. The results of NGS/WGS (next generation sequencing/whole genome shotgun) can cover more than 99% of the genetic information of the microorganism.

Whole-genome multilocus sequence typing (wgMLST) is a genotyping method in which several thousand—virtually all known—species-specific alleles are determined from the whole genome of the microorganism. In the case of *Staphylococcus aureus*, the pan-genome MLST scheme in the BioNumerics v7.6 software used in this study contains a total number of 3897 wgMLST alleles [17]. A total of 1861 comprises the core genome, an additional 2036 accessory loci, and the traditional seven MLST (multilocus sequence typing) sequences add up to complete the pan genome scheme of 3904 loci. Each wgMLST allele that is called within the genome is given a single numerical value according to its nucleotide sequence when compared with the database. If the sequence is not identical with any of the previous results, a new number is assigned. The identical number means identical nucleotide sequences on the locus, the difference of any extent is compressed into a different number for that locus. The pattern of allele numbers can be used to create clusters with a numerical representation of similarity between samples. One method is the creation of a minimum-spanning tree network. A planar graph is created from the wgMLST data. Points represent the strains, and the edges represent the genetic distance—in our case, the number of different loci, 0–3904—between each pair of isolates. The minimum-spanning tree is achieved by only keeping the shortest edge for as many points as possible. A point can have multiple connections, but only if this edge is the shortest connection for the other connected point.

In this study, we performed WGS on 50 *Staphylococcus* strains originating from Hungarian rabbit farms and compared the results with previous sequencing projects on the pathogen, one of our own, and 12 from other authors. This research aimed at revealing genome-wide genetical differences among strains representing the three virulence types (HV, aHV and LV, respectively) and to map the possible genetic relations between them. In the case of strains, which were identified only as *Staphylococcus* sp. before, the aim was to identify them more precisely.

## 2. Materials and Methods

### 2.1. Bacterial Strains

The bacterial strains used for the sequencing were isolated from deceased animals turned in by the farmers for necropsy and subsequent diagnostic investigations. No live animals were involved in this study. We reported the results of a large-scale survey in 2016, where we analysed the virulence type of 374 isolates [6]. We selected 15 HV, 15 aHV, 12 LV and 8 *Staphylococcus* sp., a total of 50 isolates from our archive for WGS. The 16th HV isolate was kindly provided by Katleen Hermans (Ghent University, Ghent, Belgium) as a PCR positive control. This HV strain labelled Sp17 (Spain 17) shows all the properties determined typical of this genotype in previous studies: mixed biotype CV-C (crystal violet type C) [11], sensitive to phages of phage group II (3A, 3C, and 71), shows the multiplex PCR pattern specific for highly virulent *S. aureus* strains [4], and has pulsed-field gel electrophoresis type N2, and *Staphylococcus aureus* protein A (*spa*) type t645 [5]. The draft genome sequence of SP17 was published by our research group previously as National Center for Biotechnology Information (NCBI) GenBank LBCS00000000 [18]. Digital archives provided another 12 rabbit-related *Staphylococcus aureus* genome sequences [8,9]. Detailed information about the *Staphylococcus aureus* strains used in this study is presented in Table 1.

### 2.2. Isolation of Strains, DNA and Library Preparation

The strains were isolated from the infected organs using standard methods [19]. The isolates that were identified as *Staphylococcus* sp. had the following characteristics: Gram-positive, catalase-positive, oxidase-negative cocci with clustered aggregations, forming medium-sized, haemolysing, yellow or greyish pigment-producing colonies. *Staphylococcus aureus* species and virulence type were determined with multiplex PCR [4]. Strains were archived at −80 °C using standard methods [20].

Genomic DNA from the *Staphylococcus* cultures was isolated using the NucleoSpin Microbial DNA Kit (Macherey-Nagel, Dueren, Germany) according to the manufacturer’s instructions. The quality and quantity of the isolated DNA was assessed by measurements using a Qubit 4.0 fluorometer (Invitrogen, Waltham, MA, USA) and Tapestation 4150 systems (Agilent, Santa Clara, CA, USA).

The NGS libraries were prepared using the Nextera DNA Flex Library Prep Kit (Illumina, Eindhoven, The Netherlands) with Nextera DNA CD Indexes. The NGS libraries were sequenced on an Illumina MiSeq instrument using the MiSeq Reagent Kit v3 using paired-end 300 bp reads. Libraries were scaled to exhibit at least 100×. All raw sequencing data are available in the European Nucleotide Archive (ENA) under the project number PRJEB37661 and ERR4017499-ERR4017552 individual accession numbers.

### 2.3. Traditional Genotyping

The traditional MLST for *Staphylococcus aureus* targets seven house-keeping genes. Regions within *arcC* (Carbamate kinase), *aroE* (Shikimate dehydrogenase), *glpF* (Glycerol kinase), *gmk* (Guanylate kinase), *pta* (Phosphate acetyltransferase), *tpi* (Triosephosphate isomerase), *yqiL* (Acetyle coenzyme A acetyltransferase) coding regions were amplified and determined with Sanger sequencing. The result was compared with the PubMLST database, and each locus was converted into a single numerical value to determine the MLST sequence type (ST) of the strains [21,22]. The multivariable region within the *Staphylococcus* protein A coding gene was amplified with standard primers [23]. Both the MLST and *spa* types were assigned using the Bionumerics v7.6 software.

### 2.4. Bioinformatic Analysis

The fastq files were imported directly from Illumina BaseSpace to the BioNumerics version 7.6 software’s (Applied Maths NV, Belgium) cloud-based calculation engine. De novo sequence assemblies were made with the SPAdes de novo genome assembler (version 3.7.1) [24]. The raw reads (fastq files) and the de novo assembled genome of each isolate were submitted to the BioNumerics’ *S. aureus* wgMLST scheme for assembly-free (AF) and assembly-based (AB) wgMLST allele calling. The quality of the sequence read sets, de novo assemblies, and assembly-free and assembly-based allele calls, were assessed using the quality statistics window in BioNumerics. The wgMLST scheme contained 3897 wgMLST loci and the 7 traditional MLST loci, the allele sequences were matched with a database, and each allele was translated into a single numerical value. Traditionally, 7-gene MLST typing and *spa* typing were also determined from de novo assembled genomes and were also confirmed by Sanger sequencing.

A whole-genome MLST-based minimum spanning tree graph (MST) was constructed to infer the genetic distance of the isolates. MST is a planar, edge-weighted undirected graph, which connects all the points together, without any cycles, and with the minimum possible total edge length. All the distances are calculated between points, and the shortest distance is kept for each point.

For the annotation of draft genomes, we used RASTtk [25] and similar genomes were identified using the Similar Genome Finder service of PATRIC 3.6.3 [26].

## 3. Results

The WGS produced data files with excellent quality. The quality score on all bases was 35–36, with an average of 35.57. The percentage of bases with 30 or higher quality score (Q30%) was 89–94%. Coverage was 103.0–246.0, with an average of 156.6. Reads were assembled to 25–84 contigs (median 41), the median contig size in genomes was 120–610 kbp (thousand base pair). The 2683–2862 kbp draft genomes were constructed de novo with 103–246 average coverage. Draft genome sequences will be provided on reasonable request.

The analysis of the draft genome sequences confirmed that the identification of 43 *Staphylococcus aureus* strains was correct. All HV strains belonged to the ST121 type and a t645 *spa* (*Staphylococcus* protein A) type (Table 1), as well as one isolate from Scotland, and the strains from Spain and Italy, according to previous reports about HV *Staphylococcus aureus* strains [2,5,8]. All aHV strains showed a novel MLST allele combination (deposited by the authors as ST5993 in the pubMLST database) because of unique *gmk* gene (encoding guanylate kinase) and *pta* gene (encoding phosphate acetyltransferase) sequences. The aHV strains were mostly *spa* type t4770 (73%), and one was the very similar t711 type. All these strains originated from a group of farms where the breeding animals were provided by the same genetic centre. Three other aHV strains (20%) had the t2407 *spa* type and these isolates were cultured from diseased rabbits from the same Ócsa production unit.

LV strains make up five different MLST patterns and seven different *spa* types. Six strains (50%) had been identified as ST2855. Three ST types and two *spa* types were not identified before. The three novel MLST sequence types, ST6063, ST6064 and ST6083, had been deposited by the authors in the pubMLST database. Two strains belonged to the very first MLST group (ST1), originating from the same isolated small-scale rural unit (Bükkösd). All the LV strains had identical *aroE* and *tpi* sequences, and *arcC* and *pta* were also homologous in all but the ST1 strains. Each of the three novel MLST sequence types (ST6063, ST6064 and ST6083) only differ by just one nucleotide from the major ST2855 type of the LV strains. The ST2855 strains comprise three different *spa* types: three t11218, two t1190 and one t4022. The ST6064 strain shared the t11218 *spa* type, and however the two other new ST types revealed novel *spa* types, but their repeats differed only by one repeat from the major *spa* type of t11218. The ST1 strains were t127 and the ST96 strain was t2802 *spa* sequence type (Table 1.).

The MST network constructed from the wgMLST analysis classified the *Staphylococcus aureus* strains from Hungary into four main clusters: HV, aHV, LV and the two ST1 isolates from Bükkösd formed a separate cluster (Figure 1). The length of the edge and the indicated number show how many loci are neighbouring strains apart, and the edge length uses logarithmic scaling of the distance. Except for the ST121 strains, the sequences derived from previous studies were very distant from the three virulent types of HV, aHV and LV. The minimum spanning tree visualization shows that the strains from previous studies, as well as the groups of the three virulence types, have a marked distance of ca. 1730–1800 allelic difference (ca. 45% of all examined loci) from the centrally located Scotland—1998 strain. This isolate showed only 150 loci difference from the Stirlingshire isolate thus these two strains could be interpreted as one group. The two Manchester strains also formed a separate cluster (Figure 1).

The ST121 strains formed a distinguished group with inner distances ranging from 4–146 allelic differences. The average pairwise distance of 12 Hungarian HV ST121 isolates was 11.5 allelic difference, which is significantly less than the proposed clonality threshold of 24 alleles for *Staphylococcus aureus* [27]. Three Hungarian HV isolates (34-Cegléd, 41-Alsótold and 42-Ócsa) were also part of the ST121 clade but showed a distance of >100 allelic difference (Figure 1).

The aHV strains form a similarly distinct cluster from the other groups. The wgMLST revealed polymorphism on an average of 21.75 loci, which represents 0.55% of the complete set of all examined genetic features. Within one *spa* type (t4770), the allelic difference was 12–29 (0.3–0.7%) alleles, while the t711 strain had 37 (0.95%) and the t2407 had 64 (1.6%) different alleles from the closest t4770 strain. LV strains grouped into two main clusters, 10 out of 12 strains clustered as a distinct group, but the distance of the individual isolates was greater than in the case of the HV and aHV isolates, which is according to the variance in the MLST and *spa* types within this virulence type. The two ST1 strains (4-Bükkösd and 5-Bükkösd) completely separated from all the strains and the clonal groups on the MST. Apart from the isolates form Bükkösd, the 9-Fülöpháza strain appears as a bit of an outlier among the LV strains with a distance of 321 alleles from the closest LV strain. The remaining nine LV isolates possess an average pairwise difference of 52.4 alleles, which is significantly more than in the case of either the HV or the aHV strains.

The genomes published by Holmes et al. [9] (except the Scotland 1999, ST121/t645 strain) were very different from the groups described above. The two strains from Manchester were closely related, as they both shared ST30/t021 types as described before. The Stirlingshire ST3092/t15410 and the Scotland 1998 ST3120/t13114 strains were different only on 150 loci (3.8%). The two strains from Glasgow and the ST3126/t1614 from England were different on 657–1800 (16.8–46.1%) loci, so they appear as individual clusters on Figure 1.

The strains classified as *Staphylococcus* sp. were isolated from specific lesions of rabbits with standard methods, but no further examinations were conducted on them until this study. These samples were not included in the MST network, as wgMLST scheme is *Staphylococcus aureus* species-specific and would have failed to identify the wgMLST loci. The PATRIC Similar Genome Finder service resulted in hits of *Staphylococcus saprophyticus*, *xylosus* and *cohnii*.

## 4. Discussion

Staphylococcosis is one of the most important infectious diseases challenging rabbit farmers and veterinarians in modern commercial rabbit meat production. Virulent genotypes can cause epidemics that call for radical solutions, so genotyping the pathogen is a crucial element of diagnostics. Our earlier study showed that a previously rarely isolated, atypical variant is the most common type among the isolates collected from Hungarian rabbit farms [6]. The multiplex PCR method used to differentiate the highly virulent strains [4] was only capable of indicating that two HV specific sequences were unable to be amplified from the aHV strains. The analysis of the whole genome of the *Staphylococcus aureus* isolates showed that HV, aHV and the majority of LV strains form separate genetic clusters with a high similarity of strains within each cluster, indicated by the allelic differences that support the clonal origin of the genotypes described 10–15 years ago.

The discriminatory power of the wgMLST method is outstanding: it examines 3904 loci of the pathogen. Some strains in this survey showed genetic polymorphism on more than 3500 loci compared to the centrally located strain with the largest distance from all clusters, and even within the same MLST ST group, hundreds of loci showed polymorphism, which can facilitate a very fine differentiation of isolates. The multiplex PCR method published in 2007 is reinforced by the results, since it can clearly differentiate high and low virulent isolates [4].

All ST121 HV strains formed a closely related cluster, and the Hungarian strains were very closely related to the strains originating from commercial producing units in Italy and Spain, and the strain isolated from a companion rabbit in 1999, in Scotland. *Staphylococcus aureus* ST121 is a globally disseminated hypervirulent clone, important in human medicine on every populated continent [28]. The clonal origin of rabbit HV strains was confirmed decades ago [5] and the significant pathogenicity of this genotype was experimentally proven [29,30]. It is clearly demonstrated that the infections on Hungarian rabbit farms are part of an epidemic spreading worldwide. Most European rabbit meat producers choose from a few hybrid breeds of rabbits, so it is not surprising that similar virulent pathogens could appear in distant countries. Most of our HV isolates originated from farms that are integrated in the production line of a rabbit meat company. The integrator has been providing breeding animals for the farms and most of them rely exclusively on this source for animals. Interestingly, one Hungarian HV isolate (42-Ócsa) is placed outside the other Hungarian ST121 isolate cluster (Figure 1.). The production unit of its origin had no contact with the aforementioned integration, and thus was able to sustain an independent supply of breeding animals. This finding demonstrated that ST121 HV strains found multiple ways to contaminate commercial rabbit production in Hungary.

The aHV genotypes are almost as closely related to each other as the strains of the HV cluster. It is well known that in closed, industrial rabbit populations, different *Staphylococcus* variants can cause health problems [2,5,6]. This genotype was detected in 1994 on four Belgian farms, where the rabbits originated from the Czech Republic [10]. In this case, the solution for the epidemic was the complete eradication of the affected populations, and since then this genotype was not reported until a Hungarian survey in 2014 [6,10]. Our results clarified that these isolates, until now only identified by the amplification of a few selected genetic elements, constitute a group of closely related, most probably clonal specimens of *Staphylococcus aureus*.

The aHV group presented a novel MLST sequence type (ST 5993). Besides the unique *gmk* and *pta* sequences, other ST alleles were also very rare variants. At the time of the study, 5993 *S. aureus* MLST profiles are listed in the PubMLST database, and among them only four and three contain the same allelic variant of *arcC* and *glpF*, respectively [21].

The whole genome analysis of the LV strains also provided interesting new discoveries. LV strains were considered to be less pathogenic variants based on biochemical properties related to human- or poultry-associated strains. The minor cluster confirms this idea, but the major cluster suggests that these strains might represent a third rabbit-associated clonal lineage. The similarity between the three main clusters is very low, around 22.8% of all loci have the same allele between the aHV and LV strains, and only 6.4% between the HV–aHV or HV–LV genotypes. The majority of the strains formed a cluster of very closely related strains, despite the fact that both MLST and *spa* sequence typing methods resulted in great diversity. The traditional seven-gene MLST and the single *spa* sequence would have distributed these 10 LV strains into seven different clusters, while the analysis of 3897 loci revealed that within the main cluster, only an average of 1.3% of the genetic features was different. Based on the wgMLST, the clonal origin of this cluster also seems very probable. The origin of the outlier ST96/t2802 LV strain from Fülöpháza remains unclear. The difference in MLST and *spa* coding sequences is only a few nucleotides, but the wgMLST showing four times more different loci than the average between other members of the LV group reinforced the distinctiveness of this variant. Further sequencing projects on LV strains might reveal other specimens of this sequence type.

The second branch within the LV genotype only contained two strains, isolated in the same diagnostic case from two different animals. This rabbitry is a small-scale rural unit, where technological problems and poor hygiene could have contributed to the infection of the animals. ST1 is an important variant of this pathogen in both human and veterinary medicine. The clonal lineage has low host specificity, thus the presence of such strains may imply human-originated contamination or poses a risk for the zoonotic transmission of the bacteria.

Attili et al. examined the genotype of 96 *S. aureus* strains, all isolated at the same farm, but from both animals and farm workers. All of the strains were classified as LV, but similarly diverse, five different *spa* types were detected among these LV strains (t094, t491, t605, t2036, t2802), none of which were similar to the *spa* type of the strains from Hungary [3].

The different virulence types can coexist within the same farm, but they have never been isolated simultaneously from the same animal in our studies [6]. The result of whole-genome sequencing confirmed that these variants are different clonal types indeed, the strains within a virulence group are similar to each other, and very different from other genotypes presenting at the same time in the same production unit. This reinforces our previous finding about the coexistence of different virulence types of *Staphylococcus aureus* on rabbit farms. Microbes of the skin and mucosal microbiota compete with each other for space and resources and the interference between the different variants of the same pathogenic species would be an interesting topic for future research. Some variants could be less virulent but more successful in colonisation, and such strains could be useful to decrease the spread of more aggressive variants.

The WGS sequencing of *Staphylococcus* strains, which did not contain the *femA* gene, thus classified as *Staphylococcus* sp. before, promised the possibility of identifying new pathogenic species related to rabbit staphylococcosis [6]. However, the results showed that these strains were closely related to bacteria, which are members of the dermal microbiota, but are also known for their possible etiological role in dermatitis [31]. All of these strains were isolated from contaminated skin lesions, and their presence might be secondary to physical trauma of the dermis, but also could be the origin of the disease. Animal experimental models [29,30] could decide this question, however, neither the negligible prevalence, nor the clinical importance of this infection could justify such research. Despite the questionable clinical importance, the results and the sequence data can be useful for further studies.

## 5. Conclusions

Next generation sequencing is clearly the future way of diagnostics and epidemiology. WGS in this project was conducted at a lower expense per sample when compared with the eight PCR reactions and sequencing needed for conventional MLST and *spa* typing cost. The PCR and Sanger sequencing methods remain useful for diagnostics where NGS equipment is not available, but their limitations are clearly demonstrated with the LV strains in this study. These strains are classified as several different MLST and *spa* sequence types, while whole-genome multilocus sequence typing revealed that these strains are very closely related.

Our results confirmed that several different genotypes of *Staphylococcus aureus* can cause clinical problems in commercial rabbit farms at the same time. Diagnostic procedures should always determine the genotype of the pathogen, preferably multiple isolates within a case, since differences in virulence types could determine radically different treatment and prevention strategies. Quarantining and monitoring imported breeding animals from new sources are the only actions that could prevent the contamination of the whole population with perilous pathogenic agents.

## Figures and Tables

**Figure 1 animals-10-01128-f001:**
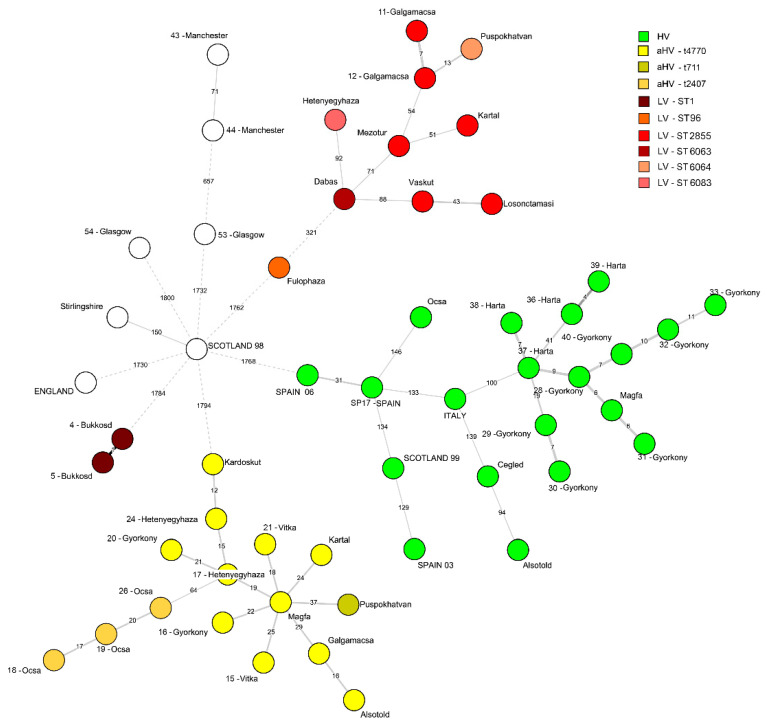
A minimum-spanning tree based on the whole-genome multilocus sequence typing profiles of 54 *Staphylococcus aureus* isolates of rabbit origin. The minimum spanning tree graph (MST) was created with the BioNumerics v7.6 advanced cluster analysis tool, where branch labels represent the allelic distances shown on a logarithmic scale. The corresponding multilocus sequence typing (MLST) and *spa* types within a virulence cluster are indicated by different shades of colours. The strains are identified with the city or country (all capital letters) of origin. Isolates are numbered as indicated in Table 1. In case there were multiple isolates from one country, the year or year/number is also provided.

**Table 1 animals-10-01128-t001:** Epidemiological and genotypic details of the *Staphylococcus aureus* isolates of rabbit origin investigated with whole-genome sequencing. Geographic place of origin is provided, where only the country is available, it is in all capital letters. The virulence type is only provided for the strains collected by the authors. Vir.: virulence type determined by multiplex PCR. Year: year of isolation of the strain.

	Accession	Year	Vir.	Origin	*spa* Type	*spa* Repeat Succession	pubST	arcC	aroE	glpF	gmk	pta	tpi	yqiL
1	ERS4418242	2009	LV	Dabas	Unknown	07-23-21-17-34-34-34-34-34-34-34	6063	12	1	611	15	11	1	337
2	ERS4418234	2010	LV	Vaskút	t1190	07-23-21-17-34-34-34-34-33-34	2855	12	1	1	15	11	1	337
3	ERS4418241	2009	LV	Hetényegyháza	Unknown	07-23-21-17-34-34-34-34-34-34-34-33-34	6083	12	1	1	480	11	1	337
4	ERS4418236	2010	LV	Bükkösd	t127	07-23-21-16-34-33-13	1	1	1	1	1	1	1	1
5	ERS4418237	2010	LV	Bükkösd	t127	07-23-21-16-34-33-13	1	1	1	1	1	1	1	1
6	ERS4418240	2009	LV	Mezőtúr	t4022	07-23-21-17-34-33-34	2855	12	1	1	15	11	1	337
7	ERS4418235	2014	LV	Losonctamási	t1190	07-23-21-17-34-34-34-34-33-34	2855	12	1	1	15	11	1	337
8	ERS4418230	2009	LV	Kartal	t11218	07-23-21-17-34-34-34-34-34-34-33-34	2855	12	1	1	15	11	1	337
9	ERS4418238	2011	LV	Fülöpháza	t2802	07-23-21-17-34-34-34-33-34	96	12	1	1	15	11	1	40
10	ERS4418233	2012	LV	Püspökhatvan	t11218	07-23-21-17-34-34-34-34-34-34-33-34	6064	12	1	68	15	11	1	337
11	ERS4418231	2011	LV	Galgamácsa	t11218	07-23-21-17-34-34-34-34-34-34-33-34	2855	12	1	1	15	11	1	337
12	ERS4418232	2011	LV	Galgamácsa	t11218	07-23-21-17-34-34-34-34-34-34-33-34	2855	12	1	1	15	11	1	337
13	ERS4418208	2013	aHV	Kardoskút	t4770	04-12-21-17-34-24-34-22-25	5993	26	3	57	478	734	4	3
14	ERS4418206	2012	aHV	Magfa	t4770	04-12-21-17-34-24-34-22-25	5993	26	3	57	478	734	4	3
15	ERS4418201	2012	aHV	Vitka	t4770	04-12-21-17-34-24-34-22-25	5993	26	3	57	478	734	4	3
16	ERS4418211	2012	aHV	Györköny	t4770	04-12-21-17-34-24-34-22-25	5993	26	3	57	478	734	4	3
17	ERS4418200	2009	aHV	Hetényegyháza	t4770	04-12-21-17-34-24-34-22-25	5993	26	3	57	478	734	4	3
18	ERS4418197	2013	aHV	Ócsa	t2407	11-12-12-12-21-17-34-24-34-22-25	5993	26	3	57	478	734	4	3
19	ERS4418198	2013	aHV	Ócsa	t2407	11-12-12-12-21-17-34-24-34-22-25	5993	26	3	57	478	734	4	3
20	ERS4418209	2013	aHV	Györköny	t4770	04-12-21-17-34-24-34-22-25	5993	26	3	57	478	734	4	3
21	ERS4418207	2012	aHV	Vitka	t4770	04-12-21-17-34-24-34-22-25	5993	26	3	57	478	734	4	3
22	ERS4418202	2011	aHV	Kartal	t4770	04-12-21-17-34-24-34-22-25	5993	26	3	57	478	734	4	3
23	ERS4418210	2013	aHV	Alsótold	t4770	04-12-21-17-34-24-34-22-25	5993	26	3	57	478	734	4	3
24	ERS4418203	2011	aHV	Hetényegyháza	t4770	04-12-21-17-34-24-34-22-25	5993	26	3	57	478	734	4	3
25	ERS4418212	2012	aHV	Püspökhatvan	t711	04-21-17-34-24-34-22-25	5993	26	3	57	478	734	4	3
26	ERS4418199	2013	aHV	Ócsa	t2407	11-12-12-12-21-17-34-24-34-22-25	5993	26	3	57	478	734	4	3
27	ERS4418205	2011	aHV	Galgamácsa	t4770	04-12-21-17-34-24-34-22-25	5993	26	3	57	478	734	4	3
28	ERS4418221	2013	HV	Györköny	t645	14-44-13-12-17-23-18-17	121	6	5	6	2	7	14	5
29	ERS4418222	2013	HV	Györköny	t645	14-44-13-12-17-23-18-17	121	6	5	6	2	7	14	5
30	ERS4418227	2013	HV	Györköny	t645	14-44-13-12-17-23-18-17	121	6	5	6	2	7	14	5
31	ERS4418228	2013	HV	Györköny	t645	14-44-13-12-17-23-18-17	121	6	5	6	2	7	14	5
32	ERS4418224	2014	HV	Györköny	t645	14-44-13-12-17-23-18-17	121	6	5	6	2	7	14	5
33	ERS4418225	2014	HV	Györköny	t645	14-44-13-12-17-23-18-17	121	6	5	6	2	7	14	5
34	ERS4418214	2009	HV	Cegléd	t645	14-44-13-12-17-23-18-17	121	6	5	6	2	7	14	5
35	ERS4418219	2012	HV	Magfa	t645	14-44-13-12-17-23-18-17	121	6	5	6	2	7	14	5
36	ERS4418216	2011	HV	Harta	t645	14-44-13-12-17-23-18-17	121	6	5	6	2	7	14	5
37	ERS4418217	2011	HV	Harta	t645	14-44-13-12-17-23-18-17	121	6	5	6	2	7	14	5
38	ERS4418218	2011	HV	Harta	t645	14-44-13-12-17-23-18-17	121	6	5	6	2	7	14	5
39	ERS4418226	2011	HV	Harta	t645	14-44-13-12-17-23-18-17	121	6	5	6	2	7	14	5
40	ERS4418220	2012	HV	Györköny	t645	14-44-13-12-17-23-18-17	121	6	5	6	2	7	14	5
41	ERS4418223	2013	HV	Alsótold	t645	14-44-13-12-17-23-18-17	121	6	5	6	2	7	14	5
42	ERS4418215	2010	HV	Ócsa	t645	14-44-13-12-17-23-18-17	121	6	5	6	2	7	14	5
43	LBCS00000000	2004	HV	SP17-SPAIN	t645	14-44-13-12-17-23-18-17	121	6	5	6	2	7	14	5
44	ERR387096	2013	-	Manchester	t021	15-12-16-02-16-02-25-17-24	30	2	2	2	2	6	3	2
45	ERR387097	2013	-	Manchester	t021	15-12-16-02-16-02-25-17-24	30	2	2	2	2	6	3	2
46	ERR387166	2013	-	ENGLAND	t1614	08-16-34-24-34-34-17-17-17	3126	3	37	19	2	20	369	32
47	ERR387196	1999	-	SCOTLAND	t645	14-44-13-12-17-23-18-17	121	6	5	6	2	7	14	5
48	ERR387195	1998	-	SCOTLAND	t13114	14-44-12-23-18-17-17	3120	18	33	6	20	7	335	48
49	ERR425000	2007	-	SPAIN 06	t645	14-44-13-12-17-23-18-17	121	6	5	6	2	7	14	5
50	ERR425012	2003	-	SPAIN 03	t645	14-44-13-12-17-23-18-17	121	6	5	6	2	7	14	5
51	ERR425013	2010	-	ITALY	t645	14-44-13-12-17-23-18-17	121	6	5	6	2	7	14	5
52	ERR494744	2009	-	Stirlingshire	t15410	15-44-12-12-17-23-18-17-17-17-17-23-24	3092	18	473	6	20	7	50	48
53	ERR494745	2012	-	Glasgow	t15409	15-12-16-02-22-12-16-02-17-24	39	2	2	2	2	2	2	2
54	ERR494746	2012	-	Glasgow	t1977	26-16-28	2257	7	6	1	5	30	8	6

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
