# Peer review of "Genomic Analysis of Staphylococcus aureus Strains Originating from Hungarian Rabbit Farms Reinforce the Clonal Origin of Various Virulence Types"

_animals, 2020, doi:10.3390/ani10071128_

Round 1
Reviewer 1 Report
The manuscript titled “Genomic analysis of Staphylococcus strains originating from Hungarian rabbit farms reinforce the clonal origin of the different virulence types” is a research article on genomic analysis of Hungarian Staphylococcus aureus and not S. aureus strains isolated in rabbit farms.
Relevant and interesting data were obtained from the Authors. These data could fulfil epidemiological gaps of S. aureus infection in rabbit farms and be used for meta-analysis studies.
Knowing that the whole genome multilocus sequence spa-Typing, spa-typing and clonal complex analysis are becoming a crucial investigations for phylogenetic studies, as well as investigate if the strains are HV or LV with zoonotic potential, this research deserves to be published.
Minor suggestions are listed below:
Title
- Line 2: please, the bacterial genus must be written in italics.
Simply summary and Abstact
- Line 19: please replace “,” with “;” because the subject change from singular to plural
- Line 29: replace “rabbit” with “rabbits”
- Lines 33 and 40: please, the bacterial genus, species and subspecies have to be written in italics.
Introduction
- Line 47: please, the bacterial genus, species and subspecies have to be written in italics
- Line 74: please, for greater understanding and readability, specify in extenso before abbreviations (NGS/WGS). Thus the specification can be omitted after, at the line 81.
Results
- Line 98: please, remember to write Staphylococcus in italics. In Table 1. throughout the Species column, please, put the name of the bacterium in italics
- In table 1. There is no correspondence between the number of strains 16 HV in the table and those written in the text (15 HV). In the table is not understandable what is the HV positive control from Ghent University. In the table are listed more than 50 strains. From number 52 to 62 what strains are they? Please, make it more understandable
- Lines 158: please, the bacterial genus, species and subspecies have to be written in italics.
Discussion
- Line 203: please, place Staphylococcus in italics
- Line 210: for the percentage, replace the comma with the dot.
References
- Please follow the instructions for authors to write the references.
Author Response
Dear Reviewer,
Thank you for your detailed response to our work. The manuscript had benn thoroughly revised according to all comments.
Results section had been substantially improved.
The table and the figure had been re-edited fo more clear presentation.
Italics had been applied everywhere.
Abbreviations had been extended at the first appearance everywhere.
The number of strains was confusing, 15 HV strains had been sequenced in this project, and 1 other Sp17 was sequenced in 2015. This had been clarified in the text.
The reference list had been managed by Zotero. Some journal abbreviations are pending, since MDPI guidelines allow this.
https://www.mdpi.com/authors/references
"The Editorial Office will abbreviate those journal titles appropriately."
Reviewer 2 Report
In this manuscript, Német et al. investigate the genetic characteristics of different Staphylococcus strains in domestic rabbits from Hungarian farms. The study is based on whole genome comparative analysis for which 51 whole Staphylococcus genomes were sequenced and 12 published genomes were used. For sequencing, the authors used short read shotgun sequencing on an Illumina MiSeq. Data analysis was done using the pipeline of BioNumerics software for allele typing and estimating phylogenetic relationships. The applied methods and analyses are state-of-the-art and, to my knowledge, comparable to other recent studies of microbial diversity.
The emphasis lies on typing all known loci for Staphylococcus aureus as well as inferring the genetic differences between the several strains. The authors detected the presence of four different clusters of Staphylococcus aureus, of which two belonged to low virulent variants, one typical highly virulent, and one atypical highly virulent variant. All variants most likely have clonal origins. The atypical highly virulent variant is the most prevalent one among the sampled rabbits in Hungary.
Despite the reported findings, this paper lacks a clear presentation and terminology. The methods need improvement regarding the data analysis and phylogenetic inferences. The main result of this study is a phylogenetic network which is wrongly called “circular dendrogram” throughout the paper. A circular dendrogram is a type of phylogenetic tree depicted in circular form whereas the shown figure is not a tree but a network. The description of this network is not precise (examples below). In general, many terms are confusing and without explanation for the used abbreviations. Other times an abbreviation is used but explained only later in the text.
Further detailed points:
- Staphylococcus should be consistently italicized.
- The Simple summary, abstract, introduction and discussion all start with almost the same sentence (“Staphylococcosis is a major disease/one of the most important infectious diseases/devastating disease… “). But nowhere it is said what kind of disease it is or what makes it so severe. Also, it might make it easier for non-medical readers to call it a “staphylococcus infection”.
- The Simple summary mentions “50 strains from Hungary” (line 15) while the abstract says ”51 Staphylococcus aureus strains” (line 33-34). This should be consistent or explained.
- Abbreviations which are clarified too late or not at all:
- Line 23: “wgMLST” should be clarified before, e.g. in line 17 “whole-genome multilocus sequence typing”.
- “MLST”, not clarified at all.
- “ST”, not clarified at all.
- spa-typing
- Line 51: “PCR”, only later clarified in line 74 “polymerase chain reactions (PCR)”
- Line 74: “NGS/WGS”
- Line 161: “SAUR loci”, not clarified at all.
- Specify that spa, gmk, pta, arc and glpF are loci in the genome. They are just mentioned without any explanation. Consistent use of italics.
- The results lack basic statistics on sequencing and genome assembly.
- Line70-71: What are the limitations? It is not clear why this is negative.
- Line 72: There are very different methods for WGS and not all are affordable. Please indicate that this is referring to Illumina shotgun sequencing of small genomes.
- Line 72: "in nowadays"
- Line 77: “is are called”
- Line 79-80: It is unclear how this assignment works and should be explained more.
- Line 92-94: The description of the “typical HV strain” includes several terms and abbreviations that are not explained.
- Line 106: What is “pure culture”?
- Line 131: These are groups/clusters and not branches. The groups are not “created” by the sequences but four major groups can be identified from the sequences.
- Line 132: "The root is calculated within the dataset". Please clarify.
- Line 133-135: What is the “marked” distance and where is this indicated in the figure?
- Line 135-136: Please rephrase the expressions "blend very well" and "are very consistent". It is not clear how they refer to the figure and what that means for the results.
- Line 139: Please explain the “t645 spa type”. (Also later in line 145: "spa type t4770 … t711 type" and line 147: “t2407 spa type”.)
- Line 143: Does "the complete feature set" refer to the genome?
- In line 172-173 it says that the multiplex PCR is not sufficient but in line 179-181 the “"multiplex PCR method ... is reinforced by our results". Please clarify.
- Line 178: Please already explain in methods and show in figure what was used as root?
- Line 182-188: This paragraph is a list of sentences without connection. Please rephrase.
- Line 192: What is a “supply for genetics”? Please clarify.
- Table 1: What are the numbers after some places of origin?
- Figure 1: Please rename to phylogenetic network and specify the algorithm for estimating it.
I do not recommend this manuscript in its current state for publication in Animals.
Author Response
Dear Reviewer,
Thank you for your detailed response to our work. The manuscript had benn thoroughly revised according to all comments.
Summary had been improved.
MLST genes had been described in detail.
Abbreviations had been extended at the first appearance everywhere.
Introductions had been improved on all points mentioned.
Results section had been substantially improved.
The dendrogram name had been corrected, it was a profound mistake. The correct minimum spanning tree had been used everywhere.
More NGS statistics had been provided.
The table and the figure had been re-edited fo more clear presentation.
Italics had been applied everywhere.
Supply of genetics is a rabbit breeding company focusing on genetic development, and selling breeding animals to meat producing farmers. This has been explained in introduction.
The number of strains was confusing, 15 HV strains had been sequenced in this project, and 1 other Sp17 was sequenced in 2015. This had been clarified in the text.
Reviewer 3 Report
Peer review:
Manuscript " Genomic analysis of Staphylococcus strains originating from Hungarian rabbit farms reinforce the clonal origin of the different virulence types” (Animals 819064)
Review comments to the authors:
The research question is important. Staphylococcus causes an important infectious disease in rabbit farming. S. aureus is the main infectious zoonotic agent and this pathogen is further classified and identified with molecular techniques, such as MLST and spa typing. Isolates are also classified according the presence of virulence genes.
The whole manuscript describes a scientific study with bacterial samples from at least five different Staphylococcus species obtained in a previous article published by the same research group. It is not clear for me, but it seems they performed WGS to identify the species as well as to determine the specific molecular patterns of the S. aureus isolates (MLST and spa types). They also reported wgMLST to compare the S. aureus isolates.
In my opinion the whole manuscript has a good sampling (51 bacterial isolates) analyzed by WGS (one state-of-the-art molecular method) and with some interesting results. However the whole manuscript is poorly written (as I mentioned before, it is not even clear the main objective of the study) and with topics not properly presented. In my opinion all of them should be improved as described below.
Simple Summary / Abstract: it is not clearly defined that the study is aiming to identify some Staphylococcus species as well as further to determine the subtypes of S. aureus with WGS. Please improve it! The authors also mention 50 isolates (more 12 genomes previously sequenced) in the Simple Summary and 51 in the Abstract (more 11 genomes). Which is correct? It is necessary improve these two descriptions.
Introduction: the beginning is Ok, but the last paragraphs need to be improved. The authors should explain better the different species involved in Staphylococcosis and the diversity of molecular patterns of S. aureus strains, instead of explaining the advantages of WGS (from lines 72 to 80).
Materials and Methods: the manuscript presents a big Table 1 with several unnecessary data. Only the Staphylococcus strains from the study should be maintained. In addition the authors should describe the methods used for determination of MLST and spa types by PCR and Sanger sequencing (lines 127 and 128)..
Results: they are very summarized and not properly evaluated. First the authors could divide in two topics to explain the identification of Staphylococcus species in one and the analysis of the S. aureus strains in another. The authors could also compare better the molecular patterns (MLST, spa types and wgMLST) according geographic region and year of collection of the samples.
Discussion: first I would suggest to compare the frequency of Staphylococcus species observed in this study with other studies. Regarding the S, aureus molecular patterns (wgMLST, MLST and spa types), the authors should compare with other studies. Finally, it is not necessary to highligth the advantages of WGS. It would be more interesting to compare the WGS results with other studies.
Conclusion: it should be shorter. Most of the text comparing with other studies should be transferred to Discussion.
Author Response
Dear Reviewer,
Thank you for your detailed response to our work. The manuscript had benn thoroughly revised according to all comments.
Summary had been improved.
MLST genes had been described in detail.
Abbreviations had been extended at the first appearance everywhere.
Introductions had been improved on all points mentioned.
The aims were clarified, in case of S. aureus we wanted to classify them, in case of Stapylococcus sp. we wanted to identify them in the first place, and the sequencing result revealed that they are not suitable for the S. aureus wgMLST scheme.
Results section had been substantially improved.
The dendrogram name had been corrected, it was a profound mistake. The correct minimum spanning tree had been used everywhere.
More NGS statistics had been provided.
The table and the figure had been re-edited fo more clear presentation. The supporting informations in the table that was used in the discussion had been replaced with spa and MLST data.
Italics had been applied everywhere.
Supply of genetics is a rabbit breeding company focusing on genetic development, and selling breeding animals to meat producing farmers. This has been explained in introduction.
The number of strains was confusing, 15 HV strains had been sequenced in this project, and 1 other Sp17 was sequenced in 2015. This had been clarified in the text.
Conclusion had been trimmed down to a more focused form.
Round 2
Reviewer 2 Report
Thank you for the thorough revision.
Német et al. replied to the comments appropriately and adapted the manuscript to an acceptable extent. The quality of the manuscript has certainly increased, making it more understandable and readable. With the aim of investigating the genetic variation among Staphylococcus strains in domestic rabbits from Hungarian farms and the validity of the wgMLST approach, the methods and results are straightforward. The discussion was improved suitably. The results of this study will benefit future diagnostic methods for these bacteria.
Author Response
Dear Reviewer,
Thank you for your positive decision. This paper had been revised and corrected by several people, naturally there are points where there was no clear consensus for linguistic style or expressions, but decisions had to be made. You checked "English language and style are fine/minor spell check required", please guide us more precisely, which lines or sentences do you find necessary to change or improve?
Thanks
Zoltán
Reviewer 3 Report
Review comments to the authors:
I have already reviewed the first version of the manuscript. The whole manuscript describes a scientific study with bacterial samples from at least five different Staphylococcus species obtained in a previous article published by the same research group. It is an interesting study, but again the full description should be improved for a best understanding of the whole study.
First of all, the authors should take care with the difference of the results obtained for Staphylococcus spp and Staphylococcus aureus isolates. As most isolated and studied strains are from S. aureus, the authors could change the Title to “Genomic analysis of Staphylococcus aureus from Hungarian rabbit farms reinforces the clonal origin of the different virulence types”.
Second, the authors are now mentioning the same number of the sequenced isolates (51) and online sequences (12) in Abstract and Simple Summary as previously recommended. But they should also correct the reported number of S. aureus (42 according Table 1, right?) in the Abstract. It is reported as 51 because they erroneously included the other Staphylococcus species (lines 31 and 32).
Third, I recommended in my first revision the authors to explain better the different species involved in Staphylococcosis and the diversity of molecular patterns of S. aureus strains, instead of explaining the advantages of WGS. The authors increased the paragraphs explaining the advantages of WGS (I think it is really not necessary), becoming the Introduction too long. I think it is worse now than before...
Fourth, the authors improved the Table 1 with more technical data (spa types and MLST profiles) of S. aureus isolates. Results are also more detailed described. But I think they should explain in the beginning of the Results (I suggest in the second paragraph) that the WGS results confirmed 42 S. aureus and 8 other Staphylococcus spp. It would be also interesting to describe that it was possible to identify 4 S. saprophyticus, 2 S. xylosus and 1 S. cohnii. In addition, the information of the last paragraph of the Results (lines 243-247) could be included here. Finally, it would be also interesting to compare the occurrence of these other Staphylococcus species in a paragraph in the Discussion too.
Author Response
Dear Reviewer,
Thank you for your work and comments.
Title corrected as requested.
In the abstract the „aureus” was not correct for 51 strains indeed. 51 strains sequenced, but 50 in this project (and SP17 HV had been sequenced in 2015). 43 S. aureus, 8 non-aureus.
The paper is focused on molecular diagnostics of rabbit staphylococcosis. Studies and books describe this disease as a pathology caused by Staphylococcus aureus infection. With simple laboratory methods our Staphylococcus sp. strains were somewhat similar to other Staphylococcus aureus isolates. All strains were isolated from specific lesions of rabbits. Thorough biochemical characterization was not done, and the multiplex PCR did not amplify femA, which is a Staphylococcus aureus specific gene. We sequenced these strains out of curiosity, and with the promise of extending the definition of rabbit staphylococcosis with other possible Staphylococcus species (as stated in lines 117-118, 344-352). The results about "non-aureus" isolates divided the colleagues when we drafted the study. Some argued that they should be left out. Finally, all authors agreed with me, that the sequence data, and the short description could be useful for other practitioners or researchers and should not be omitted. As we stated in the paper, the identified species are commensal members of the normal dermal microbiota, their presence does not mean they have notable pathogenic properties, this could be the result of contamination of physical wounds. Further investigations about these strains could be conducted later, but we decided to publicise the genomes this time, and describe this topic within this paper very briefly. The other two reviewers accepted this as well.
The description of WGS was asked by our students, who were asked to comment the paper. Molecular diversity of Staphylococcus aureus is an interesting topic, but this paper tries to focus on the diagnostics of rabbit staphylococcosis, which is very poorly described in literature. The only sequencing project we found was Holmes et al. 2016. Other studies focused on HV strains, which are the same in all countries. If you insist to describe MLST and spa results of S. aureus isolates originating from other host species I can do it. If you can only approve publication by removing the description of the methodology, I will do it, but if possible, I would prefer to keep it as is.
I added a sentence about confirmation of S. aureus identification to the results. I would like to keep the result of Staph. sp. strains at the end of the results, since it is only a minor, secondary finding. The discussion has a few sentences about these strains (lines 345-353), with references to other studies. Previous research on rabbit staphylococcosis always focused on S. aureus infection, other species are not mentioned. If you have some sources I failed to find, please cite, and I will add the comparison to the paper.